

# GeNET: a web application to explore and share Gene Co-expression Network Analysis data

Amit P. Desai[1], Mehdi Razeghin[1], Oscar Meruvia-Pastor[1,2] and Lourdes Peña-Castillo[1,3]

[1] Department of Computer Science, Memorial University of Newfoundland, St. John's, Canada
[2] Office of the Dean of Science, Memorial University of Newfoundland, St. John's, Canada
[3] Department of Biology, Memorial University of Newfoundland, St. John's, Canada

## ABSTRACT

Gene Co-expression Network Analysis (GCNA) is a popular approach to analyze a collection of gene expression profiles. GCNA yields an assignment of genes to gene co-expression modules, a list of gene sets statistically over-represented in these modules, and a gene-to-gene network. There are several computer programs for gene-to-gene network visualization, but these programs have limitations in terms of integrating all the data generated by a GCNA and making these data available online. To facilitate sharing and study of GCNA data, we developed GeNET. For researchers interested in sharing their GCNA data, GeNET provides a convenient interface to upload their data and automatically make it accessible to the public through an online server. For researchers interested in exploring GCNA data published by others, GeNET provides an intuitive online tool to interactively explore GCNA data by genes, gene sets or modules. In addition, GeNET allows users to download all or part of the published data for further computational analysis. To demonstrate the applicability of GeNET, we imported three published GCNA datasets, the largest of which consists of roughly 17,000 genes and 200 conditions. GeNET is available at bengi.cs.mun.ca/genet.

Corresponding author
Lourdes Peña-Castillo,
lourdes@mun.ca

## INTRODUCTION

Gene co-expression network analysis (GCNA) is a widely-used tool for the analysis of transcriptional profiles and a source of functional annotations for uncharacterized genes, as GCNA data is used to obtain insights on the mechanisms underlying the biological processes under study (*Filteau et al., 2013*; *Gaiteri et al., 2014*; *Parikshak, Gandal & Geschwind, 2015*). Usually GCNA's workflow involves obtaining gene-to-gene co-expression relationships from transcriptomic data (i.e., DNA microarray or RNA-seq), identification of groups of tightly connected genes (i.e., modules), and functional annotation of these modules based on gene set enrichment analysis.

There are several computer programs for gene co-expression network visualization. These programs are described in reviews by *Provart (2012)* and *Moreira-Filho et al. (2014)*. These network visualization programs lack support for, either (a) the integration of

additional information such as gene expression patterns or functional enrichment of the modules, or (b) making the data accessible online in a way that is convenient to the wider research community. In addition, GCNA data is usually summarized with images of the co-expression network highlighting only a few modules of interest (for example, *Filteau et al. (2013)*; *Jiang et al. (2016)*). This presentation style does not facilitate further exploration of GCNA data, as other researchers are unable to interactively explore this data or easily download it for further analysis. In some occasions, web applications have been developed to support the browsing and visualization of GCNA data (for example, *Childs, Davidson & Buell, 2011*; *Obayashi et al., 2009*). While this solution works in specific cases, most researchers interested in making their own GCNA data easily accessible cannot take advantage of the computational infrastructure behind these proprietary web applications, because these online tools are limited to specific organisms and lack support for content addition by external sources.

To address these drawbacks, we developed GeNET, a web application for the distribution, visualization and exploration of GCNA data. A main advantage of GeNET is that it allows researchers to upload their own GCNA data by filling out a web-based form and uploading as few as three text files in a simple tabular format. Upon submission of the data, GeNET automatically:

1. validates the data
2. computes the correlation and adjacency matrices
3. retrieves gene functional annotations from online databases such as Pfam (*Finn et al., 2016*) and KEGG (*Kanehisa et al., 2016*)
4. performs Over-Representation Analysis (ORA) of the modules
5. creates and populates a relational database based on this data, and
6. connects this database with an online user interface designed to allow querying, visualization and download of GCNA data stored in the database.

Upon creation and validation of the corresponding database, the uploaded GCNA data is freely accessible for browsing, querying and visualizing from a gene-, module- or gene-set-perspective. In addition, GeNET allows users to download all or part of the available data.

## METHODS

### Design and workflow

GeNET was designed with a 3-tier software architecture consisting of a presentation layer responsible of the graphic user interface, a business layer responsible for answering user requests, and a data layer responsible for interacting with the relational database. GeNET was implemented using Java, Apache Tomcat and MySQL. For data processing, GeNET uses R (version 3.3.1). Additionally, GeNET's presentation layer uses Cytoscape.js (*Franz et al., 2016*), an open-source network visualization engine.

GeNET's workflow is depicted in Fig. 1. As a first step, GCNA data is provided by data contributors. Data contributors are the users interested in making their GCNA data publicly accessible online. To ensure that only peer-reviewed GCNA data is available through GeNET, the submitted GCNA data must be associated with an article published

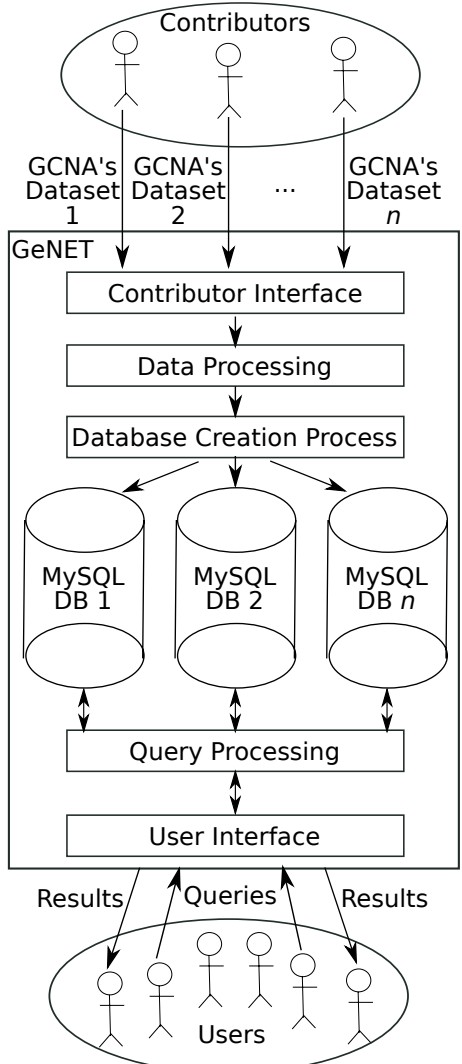

**Figure 1   GeNET workflow.**

in a journal indexed by NCBI's PubMed. GCNA data is then validated internally and, if the data passes all validation steps, the corresponding correlation and adjacency matrices are calculated, ORA is performed, and the corresponding database is created. As soon as the database is created and the data is approved by the GeNET administrators, it becomes accessible through GeNET's web site.

### Data exploration

Users can explore GCNA data from three different perspectives, which we refer to as "views": gene-centric, module-centric and gene-set-centric. In the gene-centric view, users query the database using a gene identifier. GeNET allows users to search by three types of gene identifiers, namely: gene symbols, gene systematic IDs and Entrez IDs. To simplify the search for a gene, auto-completion options are shown in a pull-down menu.
In the gene-centric view, users can see functional annotations of a gene, its expression profile across all experimental conditions in comparison with the average expression profile of the genes in the co-expression module to which that gene belongs, and the gene's neighbourhood in the gene co-expression network. In addition, GeNET links to related information available from external sources, such as NCBI, Pfam (*Finn et al., 2016*) and KEGG (*Kanehisa et al., 2016*).

In the module-centric view, users can browse the identified co-expression modules in a tabular format. For each module, GeNET provides the gene symbol of the module's hub, the number of genes in the module, and a color mark per experimental condition indicating whether average expression of the genes in that module is significantly up, or down. In this view, users can select a set of conditions, and the modules with an average expression significantly up or down in those conditions are highlighted. This helps users to focus on those modules relevant to their interests. From this tabular view of the co-expression modules, the user can navigate to a view corresponding to an individual module. In the single module view, GeNET provides a list of all the genes in the module, the expression profiles of all genes in the module across all the experimental conditions, the gene sets (pathways, protein domains or functions) over-represented among the genes in the module, and a network view of the module.

In the gene-set-centric view, users can enter a keyword, or part thereof, and query the database for gene sets whose description or identifier contains that keyword. To obtain a list of all over-represented gene sets (FDR-corrected $p$-value <0.05), the user can enter the wildcard character (*). The gene sets found are displayed in a tabular format, alongside the identifier of the enriched module, the total number of genes in the module annotated with that gene set, and the FDR-corrected $p$-value of the overrepresentation statistical test. From this tabular view, the user can click on a module and explore it using the module-centric view.

### Data access

GeNET provides the option of downloading all or part of the data for comparison or meta-analysis. To increase GeNET's integration with other available tools, the user has the option to download the correlation and the adjacency matrices as Simple Interaction Files (SIFs) to be used with a network visualization program such as Cytoscape (*Kohl, Wiese & Warscheid, 2011*).

### Data publishing

To publish GCNA data in GeNET, one simply needs to fill a web-based form (Fig. 2) and provide as few as three tabular text files, namely, a gene expression matrix, a gene information table, and a condition information table. With the information and data files provided, GeNET automatically performs module-based over-representation pathway analysis, and calculates the corresponding correlation and adjacency matrices using the R package WGCNA (*Langfelder & Horvath, 2008*) (version 1.51). Gene pair-wise correlations are calculated using either biweight midcorrelation or Pearson correlation (depending on the data contributor's input) with the complete expression profile for each pair of genes (i.e., the "use" parameter set to "pairwise.complete.obs"). The adjacency matrix contains

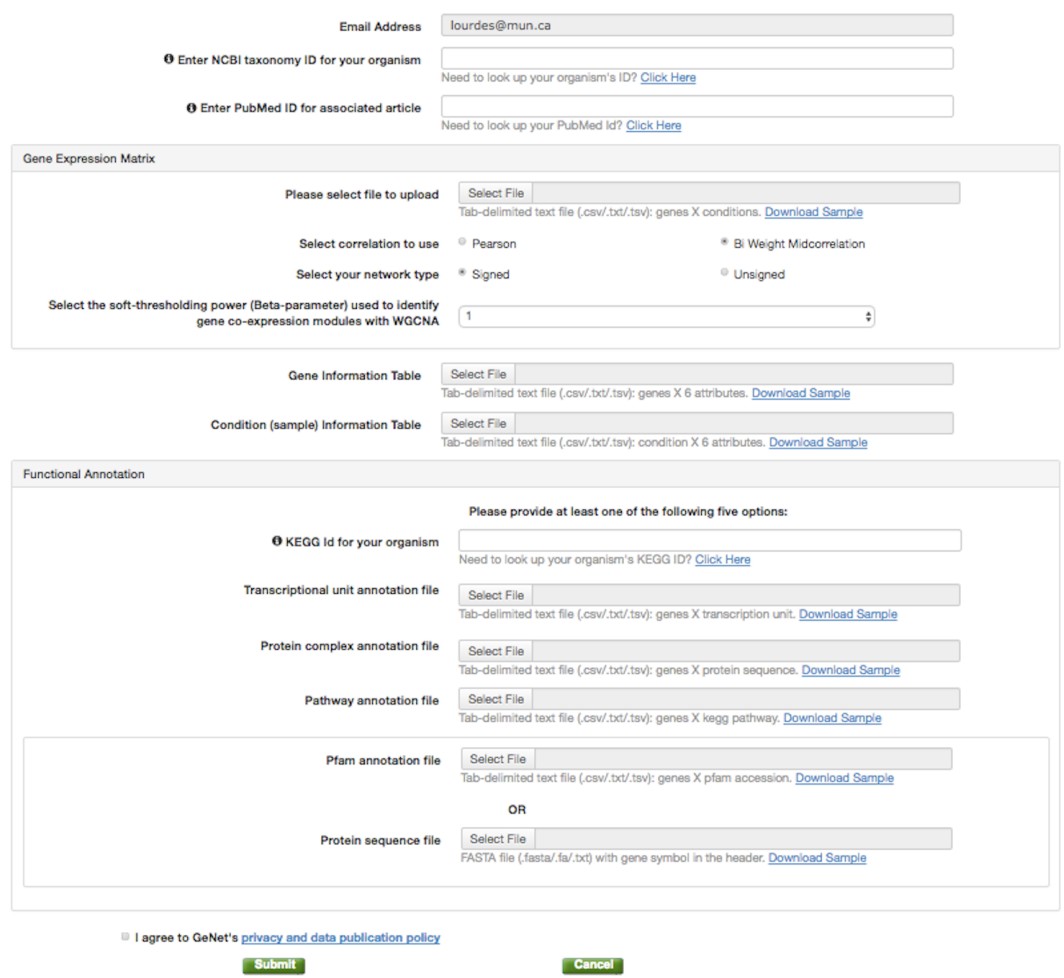

**Figure 2  GeNET data submission.** The data contributor is only required to provide three files: a gene expression matrix, a gene information table containing the genes' module assignment, and a condition information table.

a non-zero entry for all statistically significant correlations (FDR-corrected *p*-value <0.01) and is used to create the gene co-expression network. Over-representation pathway analysis is performed using the R function fisher.test with the "alternative" parameter set to "g". Additionally, GeNET automatically retrieves functional annotation from the KEGG and/or the Pfam databases using the the REST-style KEGG API (*Kyoto Encyclopedia of Genes and Genomes (KEGG), 2016*) and the RESTful Pfam interface (*European Molecular Biology Laboratory, 2016*).

Upon successful upload and approval of the data by the GeNET administrator, the GCNA data can be publicly accessed through GeNET's web interface. The detailed instructions on how to upload data are provided in GeNET's website (http://bengi.cs.mun.ca/genet/help) and in the supplementary material.

## RESULTS AND DISCUSSION

To demonstrate the functionality of GeNET, we uploaded three published GCNA data sets for:

- *Mycobacterium tuberculosis*, a pathogenic actinobacterium that causes tuberculosis;
- *Oryza sativa*, Japanese rice; and
- *Rhodobacter capsulatus*, a purple nonsulfur $\alpha$-proteobacterium.

GCNA data of *M. tuberculosis* contains expression profiles for 3,411 genes across 303 arrays, and 78 gene co-expression modules (*Jiang et al., 2016*). Japanese rice's data has expression patterns of 17,282 genes across 202 conditions and 15 gene co-expression modules (*Childs, Davidson & Buell, 2011*); whereas, *R. capsulatus*' data contains expression profiles of 3,571 genes across 23 different conditions and/or mutant strains, and 40 gene co-expression modules (*Peña-Castillo et al., 2014*). Gene expression data processing (i.e., normalization and summarization) was done for each organism as described in the corresponding GCNA publication. GCNA data for these three organisms is currently accessible through GeNET. The automatic upload exercise completed flawlessly for these data sets, indicating that GeNET is suitable for supporting GCNA data of various dimensions and able to handle GCNA data containing thousands of genes and hundreds of conditions.

### Finding modules of interest based on their gene expression in specific experimental conditions

*R. capsulatus* is an organism of interest for the production of a gene transfer agent (GTA) (*Lang, Zhaxybayeva & Beatty, 2012*). As *R. capsulatus* GCNA data includes data from the GTA overproducer strain (DE442), we used GeNET to identify modules associated with the production of GTA. To do this, we selected all experimental conditions with the DE442 strain in the tabular module-centric view (Fig. 3A). Four modules showing significant increased or decreased gene expression in the DE442 strain were then highlighted in the tabular view (Fig. 3A). Out of those four modules, the orange module was the only one with an increased gene expression specific to the DE442 strain. In GeNET's module-centric view, we examined the expression profiles of all 43 genes in the orange module, and observed that indeed their expression is increased in the GTA overproducer strain (Fig. 3B). Additionally, the orange module was enriched with gene sets representing transcriptional units containing the *R. capsulatus* GTA gene cluster (*Lang & Beatty, 2000*; *Peña-Castillo et al., 2014*) (Fig. 3C).

### Finding modules of interest based on gene-set enrichment

With an estimated 10.4 million new tuberculosis (TB) cases and 1.4 million TB deaths in 2015, the TB epidemic is larger than previously estimated (*World Health Organization, 2016*). Identifying genes potentially playing a role in the virulence of *M. tuberculosis* is a first step to characterize TB pathogenesis. We decided to use GeNET to identify modules potentially associated with *M. tuberculosis* virulence. To do this, in addition to KEGG and Pfam annotations, we uploaded into GeNET the transcriptional regulatory network of *M. tuberculosis* (*Sanz et al., 2011*) and *M. tuberculosis* Gene Ontology (GO) annotations.

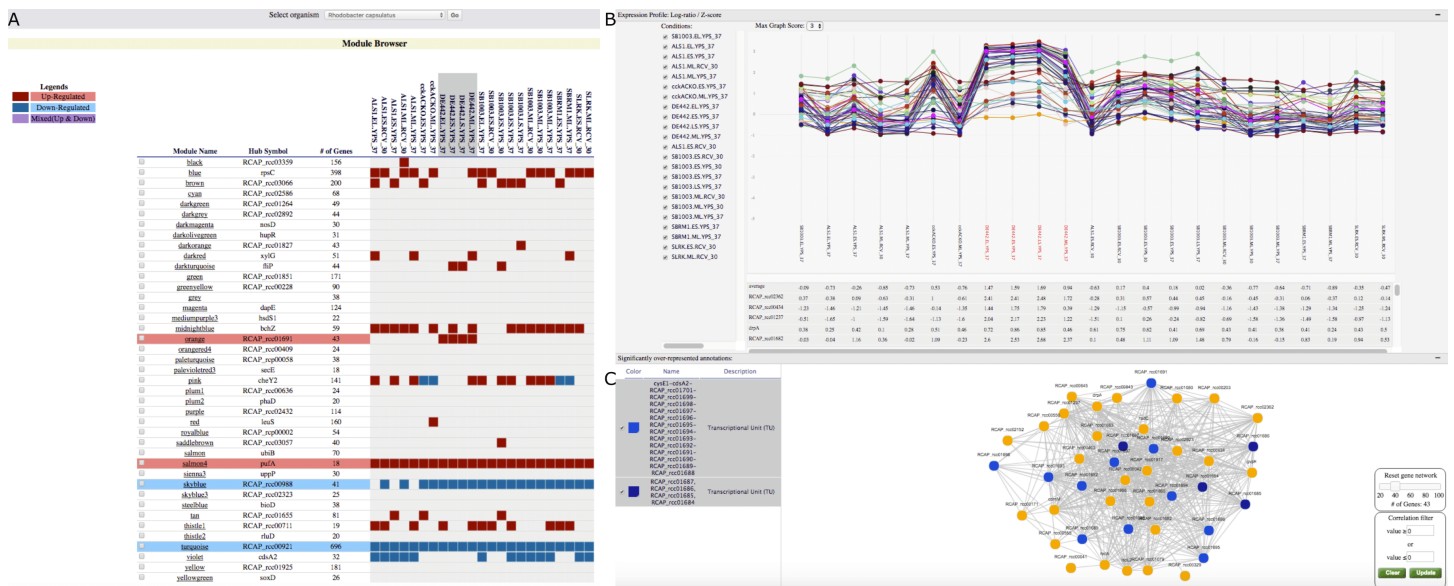

**Figure 3  GeNET module-centric view.** (A) Browsing *R. capsulatus* gene co-expression modules. By selecting experimental conditions with the DE442 strain, modules with increased (red) or decreased (blue) gene expression in these conditions are highlighted. (B)–(C) Individual module view. (B) Expression profiles of the genes in the module. (C) A network view of the module is depicted alongside gene sets over-represented among the genes in the module. Genes found in the selected enriched gene sets are highlighted.

*M. tuberculosis* GO annotations were obtained from AmiGO2 (version 2.4.24) (*Carbon et al., 2009*; *Gene Ontology Consortium, 2015*). We then searched for modules enriched with gene sets containing the keyword "host". One (the magenta module) of the two modules returned by the gene-set search was enriched with genes annotated with the GO term "growth of symbiont in host" (GO:0044117) (Fig. 4A). We further examined this module using GeNET's module-centric view, and observed that regulatory targets of transcription factors (TFs) encoded by the genes Rv0348 (*mosR*), Rv0494 and Rv0981 (*mprA*) were over-represented in this module (FDR corrected pvalues of $4.25E^{-10}, 6.07E^{-7}$, and $4.01E^{-11}$ respectively) (Fig. 4B). These three TFs have previously been implicated in regulation of hypoxia, starvation and/or virulence in *M. tuberculosis* (*Abomoelak et al., 2009*; *Bretl et al., 2012*; *Yousuf et al., 2015*). The magenta module is the only module enriched with targets of all these three TFs. The magenta module was not discussed by *Jiang et al. (2016)*, however, our analysis suggested that genes in this module might be candidates for further investigation into *M. tuberculosis* virulence.

## Visualizing expression profile of candidate housekeeping genes

For some normalization procedures, such as those performed on real-time PCR data, it is critical to have control genes displaying highly uniform expression under several different experimental conditions (*Jain et al., 2006*). These genes are often referred to as "housekeeping" genes. We used GeNET's gene-centric view to visualize the expression of a frequently used housekeeping gene in rice, tubulin beta-4 chain (also known as Os01g0805900, LOC_Os01g59150, LOC4327550) (*Jain et al., 2006*). The expression profile of this gene across 202 experimental conditions shows that indeed its expression level is

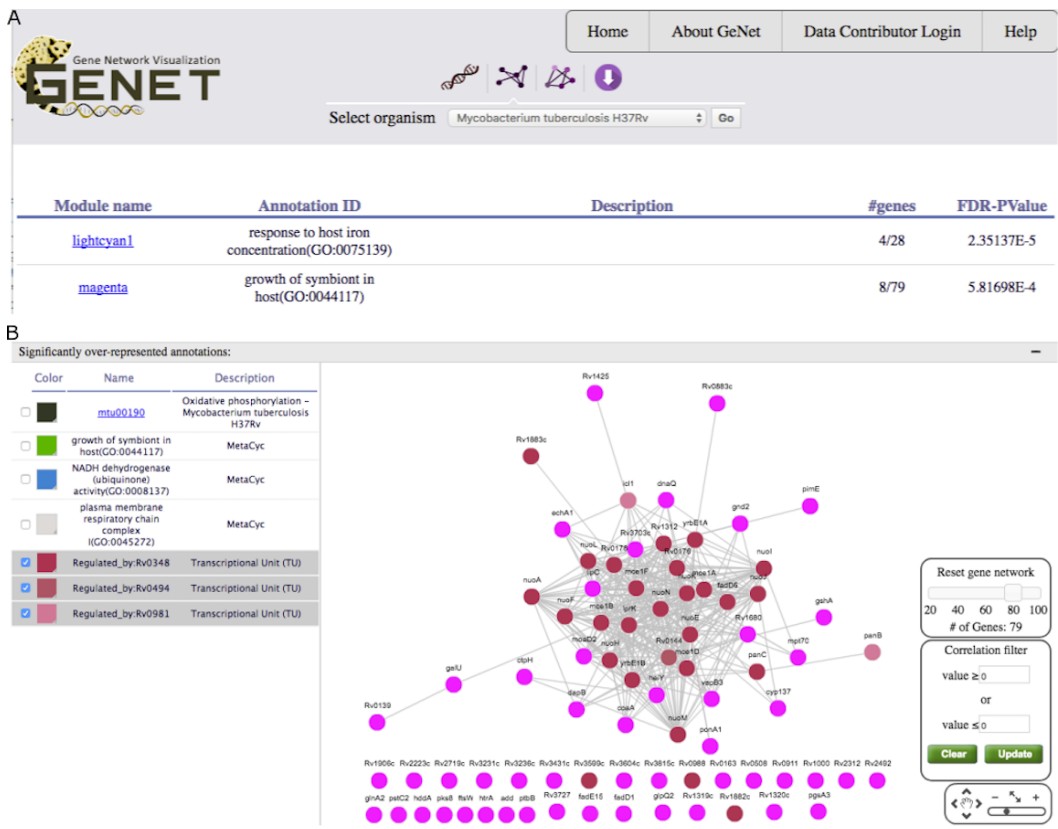

**Figure 4 GeNET gene-set-centric view.** (A) Tabular results obtained by querying GeNET for over-represented gene sets with the keyword "host" in *M. tuberculosis* GCNA data. (B) Gene co-expression network of the magenta module with the targets of Rv0348, Rv0494 and Rv0981 highlighted.

relatively constant under many experimental conditions (Fig. 5). Visualizing the expression profile of a candidate housekeeping gene facilitates confirming its suitability as a control gene and allows to identify conditions where its expression may vary. As genes belonging to the same module (blue) have similar expression profiles, one could consider other genes from the blue module to be used as control genes too.

## CONCLUSIONS

We described GeNET, an open-access online tool for publishing, browsing, and visualizing GCNA data. GeNET facilitates the process of making GCNA data available online, by providing functionality to automatically obtain most of the required data. GeNET offers a solution to integrate the diverse information contained in GCNA data and to make this information easily accessible with an intuitive, visually attractive, and user-friendly interface. Additionally, we showed its suitability to process and browse GCNA data of various dimensions, and illustrated how GeNET can facilitate the use and exploration of GCNA data by the wider scientific community.

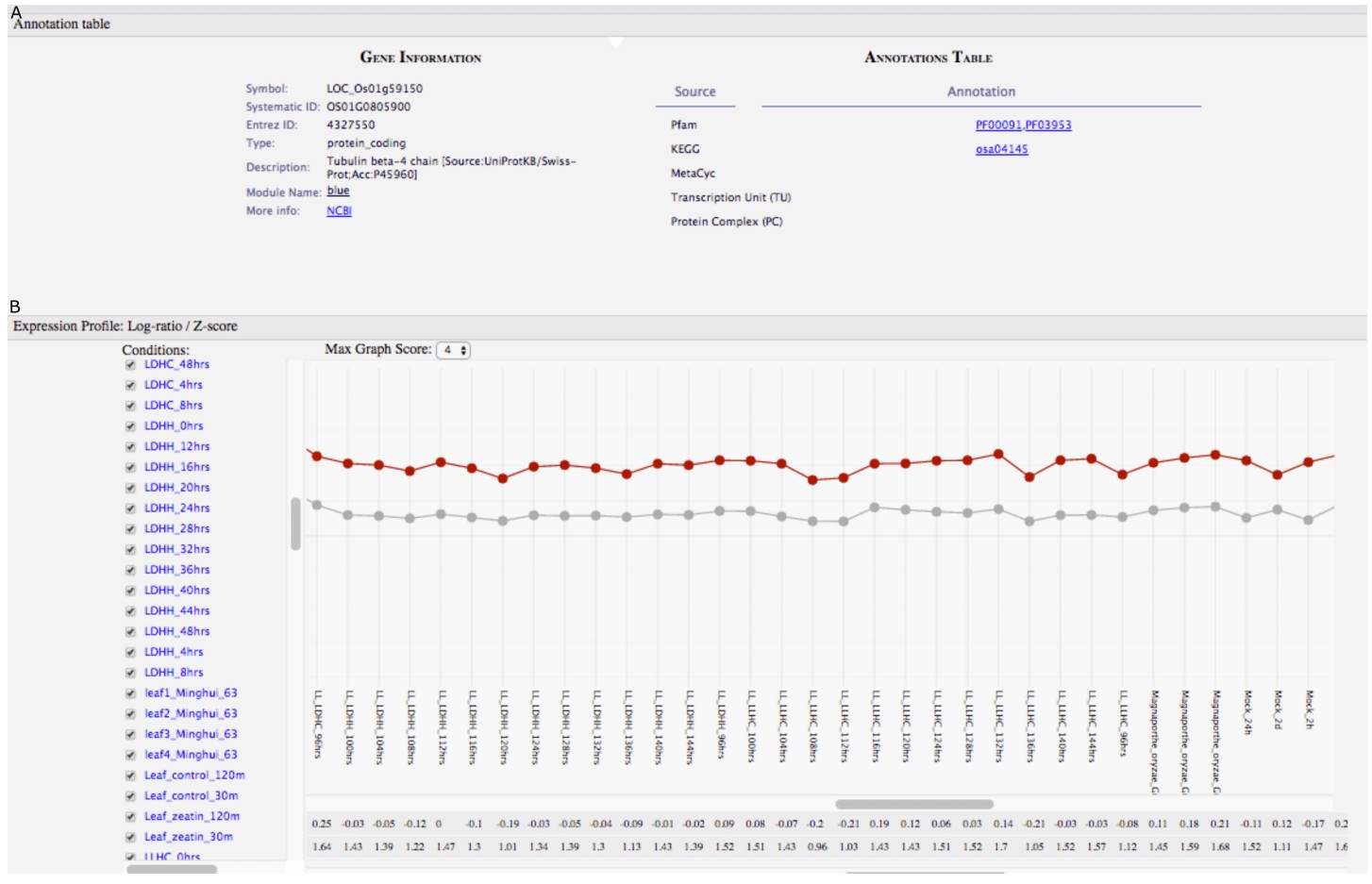

**Figure 5 GeNET gene-centric view.** (A) General information and functional annotations of the specified gene are provided. (B) The expression profile of the tubulin beta-4 chain gene (red line) is shown with respect to the average expression profile (grey line) of the genes in the blue module.

# ACKNOWLEDGEMENTS

We thank KL Childs, Ph.D., for providing the GCNA data for Japanese rice, and the editorial team at Scientific Reports for facilitating the gene expression data for *M. tuberculosis*.

## Funding

This work was supported by a Discovery Grant (No. 402087-2011) of the Natural Sciences and Engineering Research Council of Canada (NSERC) to LPC. NSERC had no role in study design, data collection and analysis, decision to publish, or preparation of the manuscript.

## Grant Disclosures

The following grant information was disclosed by the authors:
Discovery Grant: 402087-2011.

Natural Sciences and Engineering Research Council of Canada (NSERC).

## Competing Interests

The authors declare there are no competing interests.

## Author Contributions

- Amit P. Desai and Mehdi Razeghin contributed analysis tools, reviewed drafts of the paper, designed and implemented software.
- Oscar Meruvia-Pastor wrote the paper, reviewed drafts of the paper, conceived and designed software, and supervised software development.
- Lourdes Peña-Castillo analyzed the data, wrote the paper, prepared figures and/or tables, reviewed drafts of the paper, conceived and designed software, and supervised software development.

## Data Availability

GeNET is available at:

http://bengi.cs.mun.ca/genet/home.

GeNET's source code is available at:

https://amitdesai207@bitbucket.org/amitdesai207/genet.git.

## Supplemental Information

Supplemental information for this article can be found online at http://dx.doi.org/10.7717/peerj.3678#supplemental-information.

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
