# Peer review of "GeNET: a web application to explore and share Gene Co-expression Network Analysis data"

_PeerJ, doi:10.7717/peerj.3678_

## Round 0.1 · original submission · Major Revisions

· Academic Editor

Major Revisions

Our reviewers found that GeNET has its methodological novelties, but found that important details missing about the methodology. One reviewer also found difficulties in running the tool, so please take these reviews into considerations for revising the manuscript.

Reviewer 1 ·

Basic reporting

Very good.

Experimental design

Please see my review below.

Validity of the findings

GeNET works well.

Additional comments

The paper by Desai et al. presents a web application, GeNET, for the sharing, visualization, and exploration of gene co-expression data. Unlikely to other computer programs, the authors focused on making dataset accessible online in a unified procedure. GeNET also allows researchers to access all or subset of the constructed datasets. For users interested in exploring gene co-expression relationships, it provides three major functions, searching a gene, enrichment analysis for gene co-expression modules, and browsing modules. Given the advantage of such computational infrastructure, the reviewer is sure that a meta-analysis through the vast amount of publicly-available transcriptome datasets and the collected datasets here will provide a basis for future research of key pathways/processes frequently influenced by genetic- and environmental perturbations in an organisms. When accessing the website, it works well. The manuscript is generally well written but needs some attention to the items listed below before publication.

1. Detailed manual ‘how-to’
The authors should prepare the tutorial section to include actual "How-to's" on the website and Supplementary materials on the journal site. For example, identification of new genes of interest through co-expression analysis and uploading users’ own data on the GeNET.

2. The method is unclear
This reviewer is sure that co-expression approaches would be quite useful for investigating the functional genomics. However, the current website and the manuscript lacks the details information about normalization and summarization of expression data matrices. For example, did you use log-transformation of expression values? How many samples did you use for correlation calculations? The authors should clarify these points.

3. Flexibility
I have a concern about a gene/probeset identifier. For example, there are two types of gene identifiers in rice [MSU (LOC_Os IDs) and IRGSP RAP loci (OsXXg #######)]. The current GeNET does not accept RAP loci identifier. Can we use both identifiers in GeNET? Do you have any idea for a better dealing with them?

I think that these make it accessible to readers in PeerJ.

·

Basic reporting

This paper is well written and easy to follow. There is sufficient literature background to the topic and figure and tables are appropriate. The raw data is public so it doesn't have to be shared.
It presents a tool to share coexpression networks generated by the community and perform some preliminary analysis on them (such as enrichments etc...)

Experimental design

The idea of proposing a place to group all gene co-expression networks is meaningful as it is true that downloading data from the different publications is not a particularly efficient way of doing research.
Methods are described appropriately but I could not see the webpage working. I tried accessing on different days at different times and always failed. I believe I cannot judge this publication without testing how the tool provided is actually implemented and without testing it.

Validity of the findings

Without seeing the tool in action I cannot judge this paper.

Additional comments

Without seeing the tool in action I cannot judge this paper.
Please accept my apology if the error is on my part but all I get when I go to the website is the following message:
Service Unavailable

"The server is temporarily unable to service your request due to maintenance downtime or capacity problems. Please try again later.

Apache Server at bengi.cs.mun.ca Port 80"
Using chrome on a windows machine and Ubuntu as well.

---

## Round 0.2 · accepted · Accept

· Academic Editor

Accept

Both reviewers are satisfied with your revision and recommended for publication.

Reviewer 1 ·

Basic reporting

no comment

Experimental design

no comment

Validity of the findings

no comment

Additional comments

The manuscript has much improved. I appreciate that the authors took all my comments and do not have new comments.

·

Basic reporting

All requirements are met

Experimental design

All requirements are met

Validity of the findings

All requirements are met

Additional comments

It appears that the previous time I tried accessing the online tool coincided with downtime of the hosting page. I have now had a chance to use and test the tool and it seems to be working well. I really approve of the added manual on how to use it and further explanations on how the analyses are performed.
If possible I would add a set of examples of gene names to the page so that someone like me who does not know genes in any of the species could test the tool without going to look for examples. Also this would help identify the suitable IDs that should be used for the tool to work.
I am now convinced that the paper should be published and I'm sure it presents a useful resource for the community.